# Practical Stepwise Approach to Performing Neonatal Brain MR Imaging in the Research Setting

**DOI:** 10.3390/children10111759

**Published:** 2023-10-30

**Authors:** Regan King, Selma Low, Nancy Gee, Roger Wood, Bonny Hadweh, Joanne Houghton, Lara M. Leijser

**Affiliations:** 1Department of Pediatrics, Section of Neonatology, Cumming School of Medicine, University of Calgary, Calgary, 3330 Hospital Drive NW, Calgary, AB T2N 4N1, Canada; 2Department of Obstetrics & Gynecology, Cumming School of Medicine, University of Calgary, Calgary, 3330 Hospital Drive NW, Calgary, AB T2N 4N1, Canada; 3Department of Diagnostic Imaging, Alberta Children’s Hospital, Alberta Health Services, Calgary, AB T2S 3C3, Canada

**Keywords:** magnetic resonance imaging, brain, infant, safety

## Abstract

Magnetic resonance imaging (MRI) is a non-invasive imaging technique that is commonly used for the visualization of newborn infant brains, both for clinical and research purposes. One of the main challenges with scanning newborn infants, particularly when scanning without sedation in a research setting, is movement. Infant movement can affect MR image quality and therewith reliable image assessment and advanced image analysis. Applying a systematic, stepwise approach to MR scanning during the neonatal period, including the use of the feed-and-bundle technique, is effective in reducing infant motion and ensuring high-quality images. We provide recommendations for one such systematic approach, including the step-by-step preparation and infant immobilization, and highlight safety precautions to minimize any potential risks. The recommendations are primarily focused on scanning newborn infants for research purposes but may be used successfully for clinical purposes as well, granted the infant is medically stable. Using the stepwise approach in our local research setting, our success rate of acquiring high-quality, analyzable infant brain MR images during the neonatal period is as high as 91%.

## 1. Background

Imaging of a newborn infant’s brain can provide healthcare practitioners with valuable information for diagnosing potential brain lesions and maturational delays, which informs clinical decision making. Infants who are born prematurely at less than 37 weeks’ gestation are at an increased risk of neurological complications, including intraventricular and cerebellar hemorrhage (IVH and CBH, respectively), periventricular hemorrhagic infarction (PVHI), and white matter injury (WMI). In addition, preterm infants may show suboptimal brain growth and maturation [1]. Although at lower risk, term-born infants may be born with structural abnormalities of the brain or experience conditions such as meningitis, stroke, and hypoxic ischemic encephalopathy (HIE) during the perinatal and neonatal periods [1]. Accurate and early diagnosis of brain injury, infection, and/or altered development in both preterm and term-born infants can guide clinical management as well as help with the prediction of prognosis.

In addition to brain imaging in the clinical setting, increasing our understanding of preterm and term-born infant brains through research remains valuable for continued clinical care improvements. At present, the risk factors and underlying mechanisms of brain injury and altered development, and subsequent neurodevelopmental deficits, in newborn infants remain incompletely understood. These knowledge gaps may impede timely diagnosis and intervention as well as the development of new neuroprotective strategies.

The preferred, non-invasive tools to image the brain of infants during the neonatal period are cranial ultrasound (cUS) and magnetic resonance imaging (MRI). CUS is a reliable and easily repeatable bedside tool to identify the most common preterm and some term-born brain injury types and to follow the evolution of lesions (e.g., in case of WMI, PVHI, and PHVD) [2,3,4]. However, cUS is less sensitive for assessing punctate and subtle injury (e.g., punctate white matter lesions), injury to deep and superficial structures (e.g., stroke), and alterations in brain growth and maturation (including myelination and white matter tract formation), all of which may have implications for infant neurodevelopmental outcomes [3,5,6,7].

MRI has proven to be superior to cUS for assessing all brain areas, small and subtle lesions, and maturation [7,8,9,10,11]. In addition, certain MR sequences (e.g., diffusion tensor imaging, functional MRI) allow for advanced image analysis, such studying tissue volumes, white matter tract formation, and structural and functional connectivity. MRI can, however, be challenging to employ in newborn infants, particularly when the infant is unstable and/or the MRI needs to be performed urgently. Any head motion of the infant in the scanner can degrade image quality and thereby limit a reliable assessment of the brain due to incomplete visualization of lesions (e.g., small, subtle, and diffuse lesions or structural alterations). In addition, the feasibility and accuracy of an advanced brain image analyses become restricted in the case of movement artefacts. Thus, when brain MRI is performed to guide clinical care and prognostication, as well as for knowledge attainment, it is of utmost importance for the infant to remain as still as possible during the scan.

For clinical brain MR imaging, the administration of sedatives to the infant prior to scanning could be considered. However, sedation should be avoided at any cost when scanning stable newborn infants in the research setting due to the risk of side effects, including respiratory drive depression, seizures, hypotension, and agitation [12]. In addition, prolonged and/or accumulative administration of sedatives may affect healthy brain development which needs to be weighed against the potential negative effects of pain and distress [13,14,15]. Ideally, alternative, non-pharmaceutical strategies should be employed to reduce motion in infants during the MRI, such as the application of a systematic, stepwise approach to neonatal scanning including the feed-and-bundle technique.

We share our local experiences and provide recommendations for brain MRI scanning in preterm and term-born infants during the neonatal period. Our focus is on minimizing infant movement and safety to acquire high-quality MR images from medically stable, non-sedated infants for research purposes, to benefit other research groups planning to develop a neonatal brain MR imaging program.

## 2. Stepwise Approach to Neonatal Brain MRI Scanning

### 2.1. Preparation of Infant for Scanning

Neonatal brain MRI scanning for research purposes can be performed either when the infant has been discharged home or during admission to the hospital. In both cases, the arrival of the infant to the MR imaging location and scanning times are scheduled around the infant’s feeding routines, with the next feeding most optimally due just prior to the time of the scan. When scanning is performed as an outpatient, the caregivers are encouraged to arrive with their infant about one hour prior to commencing the scan to allow for sufficient time to complete the steps as outlined below. For inpatient scanning, the infant’s arrival time and first steps below may be modified on a case-by-case basis, depending on local resources (e.g., availability of support for infant preparation) and need for transportation between location of admission and MR imaging.

After the arrival of the infant and caregiver(s), the following steps are completed sequentially.

Caregivers are oriented to the diagnostic imaging department by a research team member and shown to an area which is quiet, private, and allows space for the infant to be fed, changed, and properly bundled. The team member describes the events of the visit, responding to any questions the caregivers may have and ensuring the completion of any required paperwork (such as consent forms or study questionnaires).

The infant and any accompanying caregiver are screened for MRI safety. An MRI screening form (as per local protocol) is completed by the caregiver to ensure that no MR-incompatible metal implants or devices are or will be present on or in the infant. MRI scanning relies on a strong magnetic field, and presence of metal may put the infant at risk when entering the magnetic field. The same precautions need to be taken for the caregiver should they decide to join the infant in the scanner room (see below). The information collected from the MRI screening form is carefully surveyed by the MRI technician prior to the entrance of the infant and caregiver into the scanner room.

While in the waiting area, the caregiver is encouraged to feed and hold the infant. As mentioned above, the non-sedated MRI scan is ideally planned just after a feeding time of the infant. This increases the chances that the infant takes a full feed, falls asleep prior to the scan, and will remain asleep during the scan.

After the feeding, the infant is undressed to their diaper and tightly swaddled in a blanket or hospital linen enclosing their arms and legs as a first step to reduce movement. If preferred, the infant’s t-shirt or onesie may be kept on, as long as it is metal-free (such as snaps or zippers). MiniMuffs Neonatal Noise Attenuators (Natus, San Carlos, CA, USA) are placed over the infant’s ears to protect them from the noise of the MR system that may affect the infant’s hearing (Figure 1A,B).

The above steps 1–4 provide the ideal time for the research staff to address any concerns or questions the caregivers may have regarding the MRI procedure or potential findings from the brain MRI. Caregivers may appear nervous or anxious during the visit. Typical concerns are related to how the infant will tolerate the scan or feeling strongly invested in obtaining interpretable data from the MRI scan. To mitigate any caregiver distress, we provide a careful and thorough explanation of the process that covers the extra time incorporated within the appointment to accommodate the potential needs of the infant and caregiver. We reassure caregivers that the infant will be monitored for waking or crying throughout the scan and explain that most infants will sleep comfortably for a majority of the scan. Should the infant wake up during the scan, the caregiver will be able to hold and soothe the infant.

To further minimize movement after swaddling, the infant is positioned supine on an infant-sized Medvac Vacuum Immobilization Bag (CFI Medical Solutions, Neenah, WI, USA), a tool that is effectively used for both clinical and research purposes in our institution. Other practices to reduce infant movement may be applied in other centers. The infant is positioned with their head at the top of the bundle bag. The three bottom straps on either side of the bag are fastened in the middle and the straps gently tightened around the infant’s trunk (Figure 1B). Subsequently, the air is removed from the bag by closing the secured air valve, connecting the hand pump to the bag, and evacuating the air from the bag (Figure 2A,B). The bag becomes rigidly fitted around the infant while avoiding pressure on the infant. The top portion of the bag, including a strap that is placed across the forehead and a second strap over the top of the head, are then fastened. Fastening the bundle bag may be performed immediately after feeding the infant or prior to or halfway through the feeding to prevent the infant from arousing or waking up after the feed.

Once the infant is swaddled and has fallen asleep, the team member and, if applicable, a caregiver carries the infant from the preparation area (or admit unit) to the scanner room. Prior to entering the scanner room, the infant and caregiver are again checked for any metal on or in their bodies.

If the MRI scan is performed when the infant is still admitted to the hospital, most of the infant preparation steps, such as feeding and wrapping, can take place on the unit of admission provided there are quiet surroundings.

### 2.2. Positioning of Infant and Performance of Scan

After confirming that both the infant and, if applicable, the accompanying caregiver are safe to enter the scanner room, the infant is placed supine on the MRI table with their head appropriately placed in the coil at the top of the table (Figure 2B). Infant-sized headphones may be placed over the infant’s ears, on top of the MiniMuffs, to provide additional protection for the infant’s hearing and reduce head motion (see “Safety considerations”). As alternatives to the headphones, foam padding or towels can be placed around the infant’s head, between the head and the upper sides of the vacuum bag (Figure 1B and Figure 2B). While avoiding sedation, applying an MR-compatible pulse oximetry to the infant’s foot or hand to monitor the infant’s heart rate and O_2_-saturation from the observation room is strongly recommended. Any evidence of infant distress, presenting as crying (heard through the microphone in the scanner), excessive or persistent motion, or changes in vital signs, should be evaluated immediately.

Despite the efficacy of the described preparation steps and bundling technique, movement may still occur during the scan related to waking or movement due to the noise of the MR system. As part of responsible study design, we aim to minimize the scanning time by limiting the protocol to the MR sequences required to answer the research question. In this way, important sequences may be repeated to obtain higher-quality images, when available scan time and infant comfort permit. When available, booking extra scanning time is beneficial. Dependent on the local setting, infrastructure, and patient privacy practices, caregivers may have the option to accompany the infant in the scanner room during the scan. In such cases, they can comfort the infant if they wake up during the scan by providing additional feeding, an MRI-safe soother, or sucrose drops (Sucro24, Inopro, Cowansville, QC, Canada). Additional measures that can be employed to limit infant motion and obtain high-quality images are reducing light and sound as much as possible during the scan.

### 2.3. Completion of MRI Scan and Visit

Once the MRI scan is complete, including any appropriate repetition of sequences, the infant and caregiver (if present) are guided back to the quiet area or the unit where the infant is admitted. The secured air valve of the vacuum bag can be opened, allowing air to re-enter the bag and flexibility to be restored. The straps on the vacuum bag can be unclipped, the MiniMuffs removed, and the infant extracted from the bag and bundle. At this time, the infant may be re-dressed and provided another feeding prior to completing the MRI visit. If the scan is performed when the infant is still an inpatient, re-dressing and feeding can be performed in the unit. Notably, the research team member should clearly indicate if (and when) the caregiver receives information regarding the MRI findings, subject to the local institutional and ethics policies.

### 2.4. Local MR Imaging Experiences

Using the above stepwise approach to neonatal brain MRI, our team has a continuing success rate of as high as 91% for completing the full MRI protocol consisting of eight sequences, with the acquired images being of sufficient quality for assessment and advanced image analyses. For the MR imaging to be successful, occasionally (such as in case of motion artefacts), one or more sequences must be repeated to improve image quality. As sequence repetition adds time to the MRI protocol, repetition is limited to the time available on the MRI system and, most importantly, to the infant’s tolerance. In our opinion, MRI acquisition in a newborn infant should never exceed one hour.

The reported 91% success rate is based on the experiences within our research program including MR imaging in preterm infants around term-equivalent age (40–44 weeks corrected gestational age) using the stepwise approach. In a recently completed study cohort, brain MRI was performed in 121 moderate-to-late preterm infants at a median corrected gestational age of 41.6 weeks. The brain MRI protocol consisted of eight sequences (including anatomical, diffusion, and functional sequences) with a total duration of 36 min. In 76% of the infants (*n* = 92), one or more sequences needed to be repeated to improve image quality due to motion artefacts, with the diffusion and functional sequences with longer scan durations being affected the most frequently. In these infants, the time in the MR system was extended with the scan time of the sequence(s) to be repeated; in all cases, the scan time was limited to a maximum of 60 min. In one infant, the MRI needed to be rescheduled as the infant was unable to settle, with all sequences being obtained successfully on the second attempt. In only 9% of infants, some of the eight sequences were not interpretable or analyzable due to motion corruption. While five infants (4%) in this cohort experienced a temporary increase in skin temperature (see below), no other adverse events (such as vomiting, hypoxemia, hypothermia) were encountered. Recognizing an overrepresentation of caregivers with completed higher education and professions in the medical field in the cohort, we do not anticipate that the cohort demographics will have substantially influenced our rate of successful scanning. We execute the stepwise approach similarly for all newborn infants, regardless of caregiver demographics.

## 3. Safety Considerations

MR imaging is non-invasive and safe for newborn infants if the appropriate safety precautions are taken to mitigate any possible risks. Prior to proceeding with research-related imaging, caregivers need to be provided with detailed information on the complete scanning procedure (including preparation and potential associated risks). Written caregiver consent must be obtained prior to engaging in any research activities. Regardless of the local approach to neonatal brain MR imaging, the MR safety screening forms must always be completed by trained MRI staff and research team members prior to scanning.

MR imaging is not based on ionizing or other types of radiation, like X-ray and Computed Tomography (CT), and therefore does not pose a risk of injurious exposures to the infant. MRI relies on the varying magnetization of protons in different tissues, and uses a strong magnetic field to uniformly align the protons in the target tissue, in this case, the brain. Once aligned, the scanner emits a pulse of energy (radio frequency energy) that causes the protons to re-align to their original position, emitting radio frequency energy in the process. The signals emitted by the protons are measured and used to create the image, referred to as a magnetic resonance image. Given the strong magnetic field in and around the MRI scanner, the presence of metal in the scanner room needs to be avoided at any cost. The strong magnet attracts certain metals to the scanner, putting anyone in or close to the scanner at high risk for injury and causing damage to the scanner. Hence, it is at utmost importance to confirm that any and all equipment, bundle blankets and bags, headphones, and infant clothing are free of metal and are MR-compatible prior to entering the scanner room. All individuals entering the MRI unit must be trained and certified in MR safety and adhere to the safety guidelines put in place by the local diagnostic imaging department.

The MRI system emits loud noises to which anyone undergoing an MRI or being in the scanner room during an MRI will be exposed. The noises may negatively impact hearing, particularly in newborn infants. A safety measure that therefore needs to be taken prior to MR scanning is to apply hearing protection. Specifically designed, MR-compatible ear coverings, such as MiniMuffs and infant headphones (see above), can be used for this purpose. Caregivers accompanying the infant during the MRI scan can be provided ear plugs and/or adult headphones.

### Participant-Related Considerations

When scanning for research purposes, one needs to ensure that the infant is respiratory and hemodynamically stable, not in need of intensive support, and not sick (such as with a viral or bacterial infection). Knowledge gains obtained through research should not be prioritized over the risk of scanning a medically unstable infant.

The feed-and-bundle technique using the Med-Vac Vacuum Immobilization Bag in addition to swaddling is perfectly suitable and safe to reduce infant motion during the scan. As the vacuum bag is partially open (Figure 1), the infant’s face and chest, and therewith breathing, remain visible. In the rare case of respiratory distress in a stable newborn infant, the infant’s airway is easily accessible and respiratory support can be provided. In addition, the bag can quickly be opened by releasing the clips of the straps and opening the air valve to allow air to re-enter the bag. This enables extraction of the infant from the bag within seconds in the case of respiratory or hemodynamic instability or signs of distress.

When the infant requires supplemental oxygen (without pressure or flow), such as in the case of chronic lung disease in an extremely preterm infant, the MRI can still be performed, provided hospital oxygen and the appropriate connector and tubing are available. Of note, it is of utmost importance to avoid entering the magnetic field with a non-MR-compatible oxygen cylinder. Additionally, orogastric or nasogastric tube feeding, previous ligation of a patent ductus arteriosus or abdominal surgery, or in situ ventricular cerebrospinal fluid drainage device, are themselves not contra-indications for MR scanning, as nowadays MR-compatible devices are primarily used. However, the presence of an experienced MR technologist during the scan, preferably with experience in scanning newborn infants, is strongly recommended in these cases.

Newborn infants, particularly preterm infants scanned prior to the expected delivery date and low weight infants, are prone to temperature instability due to the limited ability to regulate and maintain their body temperature [16]. In addition, the magnetic field in MR scanning may induce heating of the person in the scanner. The above put infants at risk of experiencing hypothermia or hyperthermia during the MRI, with potential negative effects on the infant’s comfort, feeding, and glucose levels. In many centers, it is therefore common practice to measure the infant’s temperature before and after the MRI. In addition to ensuring the infant did not develop hypothermia or hyperthermia during the scan, measuring the infant’s temperature ensures they do not have a fever prior to scanning. Strategies to mitigate temperature changes, particularly hypothermia, include swaddling the infant in blankets and keeping the scanner room temperature between 15 and 18 degrees Celsius. In the rare case hypothermia or hyperthermia does occur, appropriate measures should be taken to normalize the infant’s temperature. Of note, there is literature available showing that even when the infant’s skin temperature increases during the scan, in most cases, their core temperature remains thermoneutral [17,18,19].

In our example cohort of 121 moderate-to-late preterm infants who underwent brain MRI using the stepwise approach, we only encountered a change in skin temperature as measured before versus after the MRI in five (4%) infants. Their temperature after the scan had increased either to 37.5 °C or by >1 °C compared to before the scan. However, in all infants, the skin temperature normalized within several minutes of being removed from the vacuum immobilization bag, suggesting that the temperature increase may be related to the bundling and non-breathable material of the bag. None of the infants showed any other clinical signs of infection or illness, such as nasal congestion, wheezing, or sleepiness.

Other safety considerations include the cautious positioning of the infant’s head in the coil in the case of a specific anatomical deviation (e.g., large hemangioma) or device (e.g., ventricular access device), and monitoring the infant for any regurgitation or signs of discomfort during the scan. Careful positioning by means of MR-safe foam padding is used effectively to elevate the head and prevent choking. In the case of infant discomfort or waking, the scanning team and/or caregiver may soothe the infant by holding and rocking, or providing a soother or sucrose drops. Caregivers or the scanning team may also discontinue the scanning if appropriate. Any signs of infant distress including crying, persistent/erratic motion, or vital sign changes, should result in an immediate halt of the scan and evaluation by the scanning team.

Local infrastructure and resources will differ with respect to brand, type, and magnetic field strength of the MR system, availability of a quiet preparation area, and infant immobilization practices. Regardless of local specificities, the above strategies can be, and all above safety precautions, should be applied.

## 4. Ethical Concerns and Caregiver Experiences

A concern that has been raised with regard to MRI scanning of the neonatal brain, both in a clinical and a research setting, is the potential negative impact the MRI results could have on the caregivers’ mental health and well-being [20,21]. For some families, the results of the brain MRI may provide reassurance and/or answers, either by exclusion or confirmation of a brain abnormality, which may help with the prediction of prognosis and access to early intervention programs. However, for other families, further medical investigations and the potential to detect unexpected brain lesions or brain lesions that are of unclear clinical significance may cause distress or anxiety. Edwards et al. [22] systematically evaluated the well-being of mothers after receiving the results of their newborn infant’s cUS scan and after additionally receiving the results of the more detailed brain MRI. Their findings showed that maternal anxiety levels were lower after receiving the results of both the cUS and MRI compared to only receiving the cUS results [22].

The research conducted by our team involves brain MR imaging in both preterm and term-born infants during the neonatal period. In order to optimize our approaches with respect to communication style, information provision, and responding to caregiver questions when starting the imaging program in our local center, we assessed caregiver experiences with the brain MRI before and after the scan. The short survey included questions regarding the caregiver’s prior experience with MRI, concerns about their infant’s brain MRI, whether concerns were addressed appropriately by the team members before and during the MRI visit, and suggestions for improvements. The common concerns expressed by caregivers prior to the MRI scan applied to possible harmful exposure to radiation and loud noises from the MR system, and risk of distress of their infant during the scan. The caregiver feedback after scan completion, however, highlighted that caregivers felt sufficiently reassured that no radiation was emitted during the scan, felt that hearing protection was managed appropriately, and that allowing them to accompany their infant in the scanner room eased concerns of distress.

With research ongoing, caregivers regularly express their gratitude for the opportunity to learn more about their infant’s brain (particularly when they would not otherwise qualify for a clinical brain MRI), with the potential of referral of the infant to appropriate services if deemed necessary based on the MRI results. With proper information provision about MRI safety as well as informed responses to common questions about radiation and noise, caregivers are typically put at ease and enthusiastic for their infant to contribute to scientific research and advancing care and outcomes for future newborn infants.

## 5. Limitations and Future Improvements

The experiences and recommendations on the approach to neonatal brain MR imaging described in this manuscript are limited to medically stable, non-sedated newborn infants imaged during the neonatal period and up to several months post-term in a setting of a previously established research infrastructure. Brain MR imaging in unstable newborn infants, requiring additional safety considerations that bring more logistical challenges, is beyond the scope of this manuscript. As per the manufacturer’s instructions, the vacuum immobilization bag is only intended for use for newborn infants from 0 to 5 months. In addition, the high rate of success with brain MR imaging in our local setting is partly related to the availability of the MR system. The system is dedicated to research, which allows for the booking of additional scanning time and therewith repetition of sequences in cases of infant motion or waking. The use of clinical MR systems and/or differing institutional policies may inflict local time constraints.

While the described stepwise approach has shown to be effective for non-sedated neonatal brain MR imaging, improvements can be sought to further reduce infant movement and waking and therewith image motion artefacts. Studies are ongoing, particularly in view of diffusion and functional imaging, to reduce MR scan times and therewith the time infants need to remain still, while maintaining the high resolution of the images. In addition, developing MR-compatible noise cancelling equipment and a vacuum bag made of a breathable material may reduce infant waking.

## 6. Conclusions

We describe our local practices and provide recommendations for obtaining high-quality brain MR images in stable newborn infants, primarily focused on non-sedated, elective brain MRI in a research setting. Based on our local experience, applying a systematic, stepwise approach to infant MRI scanning during the neonatal period by use of the feed-and-bundle technique is highly effective in acquiring images with minimal motion. High-quality images are critical in capturing the full extent of brain injury and abnormal brain development, which is important in the early and accurate diagnosis and prognostication of newborn infants. High-quality images also contribute to increasing our knowledge on neonatal brain injury and development, and to improving neuroprotective care, through advanced image analysis to study the developing brain both on a macrostructural and microstructural level. Most importantly, the described approach is safe and well-tolerated by newborn infants, avoiding the use of sedation and any potential risks when appropriate safety precautions are employed. If the newborn infant is medically stable and the MRI is not urgent or time-sensitive for clinical indications, the recommendations can easily be applied in one’s local setting, regardless of differing MR systems and protocols, and if necessary, can be adapted based on local experience, expertise, and infrastructure.

## Figures and Tables

**Figure 1 children-10-01759-f001:**
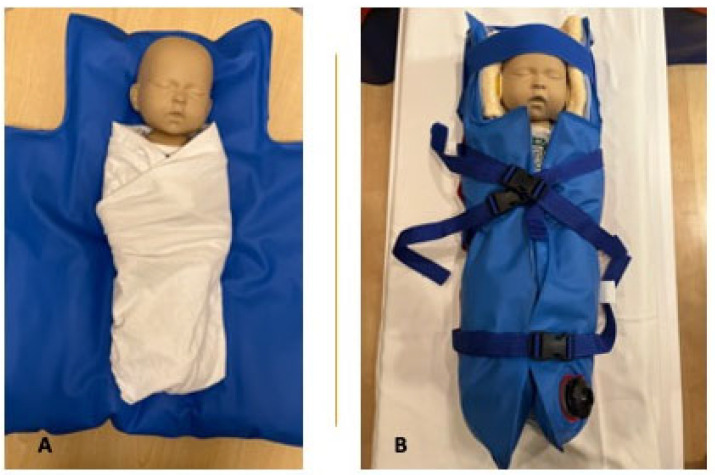
Representation of feed-and-bundle technique. (**A**) Swaddled infant prior to securing the Medvac Vacuum Immobilization Bag. (**B**) Fully bundled infant in vacuum bag.

**Figure 2 children-10-01759-f002:**
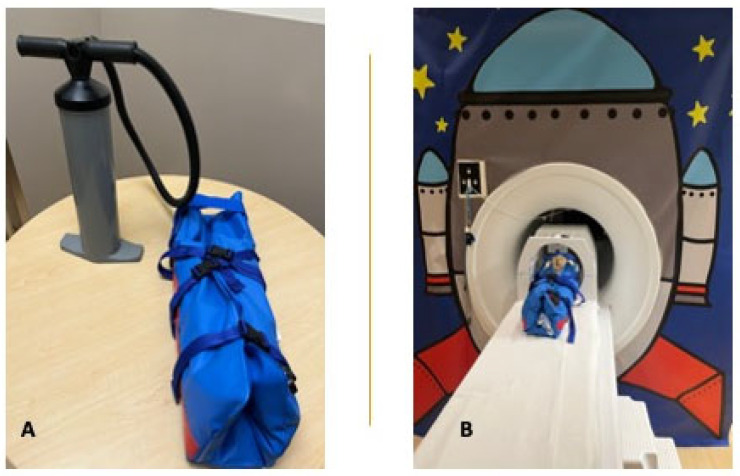
Feed-and-bundle technique in MRI scanner. (**A**) Vacuum bag and air pump required for feed-and-bundle technique. (**B**) Infant positioned in mock MRI scanner with head in coil, immobilized in vacuum bag.

## Data Availability

Not applicable; no new data were created.

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
