# Peer review of "Practical Stepwise Approach to Performing Neonatal Brain MR Imaging in the Research Setting"

_children, 2023, doi:10.3390/children10111759_

Round 1
Reviewer 1 Report
Comments and Suggestions for Authors
Dear authors, it is an interesting piece of work. However, there are a few issues to consider:
1. There is no evidence to support your results of e.g. 92% success... Please provide data on how you came up with such improvements.
2. Please provide information about your sample data and methodology applied. How many children were examined, their characteristics, etc.
3. Any limitations of your research? Is your approach applicable to all children regardless their age, size, condition, etc? Regarding the caregiver, would their experience, education level, etc affect the application of your methodology?
4. Please comment on any problems or difficulties you met during your research? If any how did you deal with?
5. Future research any suggestions?
Author Response
Authors’ reply to Reviewer 1:
Thank you very much for carefully reviewing our manuscript and for your suggestions for improvement. We address each of your comments in the point-by-point response below and have revised the manuscript accordingly. The line numbers indicated in our responses refer to the line numbers in the manuscript with the revisions, which are also highlighted in yellow.
Reviewer’s comment 1: Dear authors, it is an interesting piece of work. However, there are a few issues to consider: There is no evidence to support your results of e.g. 92% success... Please provide data on how you came up with such improvements.
Authors’ reply: The main purpose of the manuscript was to share our experiences with brain MR imaging in stable, non-sedated newborn infants, which is part of our research program on improving brain health and outcomes of vulnerable newborn infants. Based on our experiences, we provide recommendations on the use of a stepwise approach to neonatal brain MRI, to benefit other research groups that want to build a neonatal brain MR imaging program. We define successful MRI as acquiring all eight MR sequences that are part of our research MRI protocol, with the acquired images being of sufficient quality for assessment and advanced image analyses. The described rate of 91% reflects our MR imaging success rate, using the described stepwise approach, in the first (completed) cohort of newborn infants in our recently established research program in Calgary. In some infants, successful MRI entailed repetition of one or more sequences to obtain images of sufficient quality (for example for infant movement or waking). The infant cohort is described in the manuscript to show the effectiveness of the stepwise approach to imaging, rather than collecting research data from the infants in the cohort and describing the data in a manuscript. As this was the first local newborn infant cohort (with completed recruitment) undergoing MR imaging in our relatively new research program, we cannot comment on improvements in success rates.
We have added more information on the reported 91% brain MR imaging success rate and the definition of success in the manuscript (lines 205-228). In addition, we have added a heading to make more clear that the reported percentage is based on our local experiences (heading 2.4; line 205). We have corrected our previous error of 92% to 91% in view of the 9% of infants in whom less than eight sequences were obtained or were of sufficient quality.
Reviewer’s comment 2: Please provide information about your sample data and methodology applied. How many children were examined, their characteristics, etc.
Authors’ reply: As indicated in more detail above, the main purpose of the manuscript was to share local experiences and provide recommendations on the approach to brain MR imaging in non-sedated newborn infants, rather than collecting data from the infant cohort mentioned in the manuscript. We have however added more detail on the infants in the cohort and the reported brain MR imaging success rate (lines 205-228).
Reviewer’s comment 3: Any limitations of your research? Is your approach applicable to all children regardless their age, size, condition, etc? Regarding the caregiver, would their experience, education level, etc affect the application of your methodology?
Authors’ reply: Thank you for making us aware of the omission of a limitations section. We have added a paragraph on the limitations of the manuscript and the described approach to neonatal brain MR imaging at the end of the manuscript (lines 363-374; heading 5).
In addition, we have added a note on caregiver experience and education level, and their potential influence on the success of MR imaging, to the description of the infant cohort from which the described success rate was calculated (lines 231-235).
Reviewer’s comment 4: Please comment on any problems or difficulties you met during your research? If any how did you deal with?
Authors’ reply: The described stepwise approach to brain MR imaging in newborn infants builds on the experience of the senior author, Dr. Leijser, who has ample experience with neonatal brain MRI for research purposes across different centers and continents. Benefiting from the local, already established research infrastructure and availability of a research MR system, we did not experience challenges with implementing the described stepwise approach. We do, however, occasionally encounter an increase in skin temperature in infants after the MRI. In the cohort described in the paper, five infants (4%) had an increase in skin temperature. However, in all the skin temperature normalized within several minutes of being taken out of the vacuum immobilization bag. The temperature increase may be related to the bundling and non-breathable material of the vacuum bag. We have added a note on the temperature change in the manuscript (lines 229-231 and 306-313).
Reviewer’s comment 5: Future research any suggestions?
Authors’ reply: We have now added suggestions for future developments that have the potential to further increase the success rate of neonatal brain MR imaging (lines 375-381).
Reviewer 2 Report
Comments and Suggestions for Authors
CUS is the best method for evaluating newborn brains. It is widely used all over the world and it has proved to be safe and accurate.
Unfortunately sending a preterm infant for an MRI scan can be very difficult especially if the infant is intubated. Also, the number of MRI devices available for newborn scans is minimal. So, getting an MRI scan can be a mission close to impossible in a short time frame.
The sedation is also a disputed reason for not taking a routine MRI scan for preterm infants or sick infants from the NICU.
The method that you are proposing for feeding and restraining the newborn can lead to failure in a correct examination of an infant because there can still be a slight movement that can affect the image quality.
So, at least for the reasons stated above, I am not a big fan of MRI in case of emergency in neonates. I would rather choose a high-resolution CUS.
Comments on the Quality of English LanguageEnglish is fine
Author Response
Authors’ reply to Reviewer 2
Reviewer’s comment: CUS is the best method for evaluating newborn brains. It is widely used all over the world and it has proved to be safe and accurate.
Unfortunately sending a preterm infant for an MRI scan can be very difficult especially if the infant is intubated. Also, the number of MRI devices available for newborn scans is minimal. So, getting an MRI scan can be a mission close to impossible in a short time frame.
The sedation is also a disputed reason for not taking a routine MRI scan for preterm infants or sick infants from the NICU.
The method that you are proposing for feeding and restraining the newborn can lead to failure in a correct examination of an infant because there can still be a slight movement that can affect the image quality.
So, at least for the reasons stated above, I am not a big fan of MRI in case of emergency in neonates. I would rather choose a high-resolution CUS
Authors’ reply: Thank you very much for carefully reviewing our manuscript. We fully agree with the reviewer that cranial ultrasound remains the preferred tool to image the neonatal brain and, if applicable, follow the evolution of brain injury, in particular in newborn infants that are medically unstable. We describe the benefits of cranial ultrasound in the introduction of the manuscript. However, cranial ultrasound is still limited with respect to the extent of detail it provides to study the neonatal brain. To better understand the neonatal brain and related neurodevelopmental outcomes (and inform new neuroprotective strategies), we feel that the additional detail MRI provides over high-resolution cranial ultrasound adds valuable information. Also, MRI enables advanced image analyses, for example, to study structural and functional connectivity. Our manuscript and the described recommendations are specifically focused on performing brain MRI in stable, non-sedated newborn infants for research purposes; the MRIs are planned ahead of time and are not urgent or strongly time-sensitive. Brain MR imaging for clinical and/or urgent indications in unstable newborn infants, requires additional safety considerations and goes along with more logistical challenges, that need to be weighed against the information that one can get from high-resolution cranial ultrasound. The description of brain MRI in unstable newborn infants is beyond the scope of this manuscript. We have added a comment on this in the limitations section of the manuscript (lines 363-374). In addition, on several occasions throughout the manuscript we have reworded the text to make more clear that our recommendations only apply to medically stable newborn infants.
Reviewer 3 Report
Comments and Suggestions for Authors
This article introduces a stepwise approach to perform neonatal brain MRI without sedation using a vacuum immobilization bag. This is useful information as it allows us to emphasize that MRI scanning can be done safely without sedation.
Unfortunately, this article contains too little data. In this article, the data are only presented in section 2.3 that the success rate of as high as 92% and that the 8-sequence MRI protocol was scanned with a total duration of 36 minutes.
I think the following information should be presented.
Number of subjects.
Gestational age at MRI scan of the subjects.
Distribution of actual scan minutes. (both complete and incomplete scanning)
Type and frequency of adverse events. (hypoxemia, vomiting, hyperthermia, etc.)
Have there been any cases of suffering transient hypoxemia during the protocol procedure?
Authors should accurately disclose the imaging settings when titling this article "MR imaging in the research setting". Did detailed scans for research purposes (e.g. tractography) have similar success rates?
Author Response
Authors’ reply to Reviewer 3:
Thank you very much for carefully reviewing our manuscript, your positive reply and the suggestions for improvement. We address each of your comments in the point-by-point response below and have revised the manuscript accordingly. The line numbers indicated in our responses refer to the line numbers in the manuscript with the revisions, which are also highlighted in yellow.
Reviewer’s comment 1: This article introduces a stepwise approach to perform neonatal brain MRI without sedation using a vacuum immobilization bag. This is useful information as it allows us to emphasize that MRI scanning can be done safely without sedation.
Unfortunately, this article contains too little data. In this article, the data are only presented in section 2.3 that the success rate of as high as 92% and that the 8-sequence MRI protocol was scanned with a total duration of 36 minutes. I think the following information should be presented. Number of subjects. Gestational age at MRI scan of the subjects.
We have now included more information on the infant cohort, including size of the cohort and the average age at MRI, in the manuscript as suggested (lines 206-235).
Reviewer’s comment 2: Distribution of actual scan minutes. (both complete and incomplete scanning)
Authors’ reply: Thank you for making us aware of this omission. We have added the time of the full protocol and the time of the infants in the MR system in the manuscript (lines 218-224).
Reviewer’s comment 3: Type and frequency of adverse events. (hypoxemia, vomiting, hyperthermia, etc.)
Authors’ reply: We have added details on the occurrence of adverse events, consisting of a short-lasting increase in measured skin temperature in five out of the 121 infants after the MRI, to the manuscript (lines 229-231 and lines 306-313). Fortunately, we have not encountered other adverse events.
Reviewer’s comment 4: Have there been any cases of suffering transient hypoxemia during the protocol procedure?
Authors’ reply: Fortunately, we have not encountered episodes of hypoxemia in any of the infants in the described cohort (or the ongoing cohort) in our neonatal brain imaging research program. We have mentioned the experienced adverse event, that is temporary skin temperature increase, in the manuscript as indicated above.
Reviewer’s comment 5: Authors should accurately disclose the imaging settings when titling this article "MR imaging in the research setting". Did detailed scans for research purposes (e.g. tractography) have similar success rates?
Authors’ reply: We have added details on the MRI protocol to the manuscript and indicated which sequences needed to be repeated the most frequently due to infant motion (lines 218-222). In addition, given the confusion the title may cause with respect to the purpose of the manuscript, we are suggesting (to the editor) to revise the title to “Practical Stepwise Approach to Performing Neonatal Brain MR Imaging in the Research Setting”.
Reviewer 4 Report
Comments and Suggestions for Authors
I read with interest the manuscript in which the authors describe an innovative approach to the principles of imaging diagnostics of a newborn. The work is written clearly. The added value is a graphical presentation of the correct positioning of the newborn for examination. I believe that the work contributes a lot to both neonatological and radiological knowledge.
I suggest that the authors pay attention to minor stylistic and grammatical errors in the manuscript.
Comments on the Quality of English LanguageI suggest that the authors pay attention to minor stylistic and grammatical errors in the manuscript.
Author Response
Authors’ reply to Reviewer 4:
Reviewer’s comment: I read with interest the manuscript in which the authors describe an innovative approach to the principles of imaging diagnostics of a newborn. The work is written clearly. The added value is a graphical presentation of the correct positioning of the newborn for examination. I believe that the work contributes a lot to both neonatological and radiological knowledge.
I suggest that the authors pay attention to minor stylistic and grammatical errors in the manuscript.
Authors’ reply: Thank you very much for carefully reviewing our manuscript and your positive response. We have corrected a few grammatical errors and the numbering of some headings throughout the manuscript. We have highlighted the corrections in yellow.
Round 2
Reviewer 3 Report
Comments and Suggestions for Authors
This manuscript has been revised adequately.